# Oral Gel Formulation of *Cotinus coggygria* Scop. Stem Bark Extract: Development, Characterization, and Therapeutic Efficacy in a Rat Model of Aphthous Stomatitis

**DOI:** 10.3390/pharmaceutics17101293

**Published:** 2025-10-02

**Authors:** Jovana Bradic, Miona Vuletic, Vladimir Jakovljevic, Jasmina Sretenovic, Suzana Zivanovic, Marina Tomovic, Jelena Zivkovic, Aleksandar Kocovic, Nina Dragicevic

**Affiliations:** 1Department of Pharmacy, Faculty of Medical Sciences, University of Kragujevac, Svetozara Makovića 69, 34000 Kragujevac, Serbia; jovanabradickg@gmail.com (J.B.); marinapop@gmail.com (M.T.); salekkg91@gmail.com (A.K.); 2Center of Excellence for Redox Balance Research in Cardiovascular and Metabolic Disorders, Svetozara Makovića 69, 34000 Kragujevac, Serbia; drvladakgbg@yahoo.com (V.J.); drj.sretenovic@gmail.com (J.S.); 3Department of Pharmacy, Singidunum University, Danijelova 32, 11000 Belgrade, Serbia; jelenazivkovic1@yahoo.com; 4Department of Dentistry, Faculty of Medical Sciences, University of Kragujevac, Svetozara Makovića 69, 34000 Kragujevac, Serbia; miona91kg@gmail.com (M.V.); suzanazivanovic91@yahoo.com (S.Z.); 5Department of Physiology, Faculty of Medical Sciences, University of Kragujevac, Svetozara Markovica 69, 34000 Kragujevac, Serbia; 6The Institute for the Study of Medicinal Plants “Dr. Josif Pančić”, Tadeuša Košćukog 1, 11000 Belgrade, Serbia

**Keywords:** *Cotinus coggygria*, recurrent aphthous stomatitis, mucoadhesive gels, oxidative stress, ulcer healing

## Abstract

**Background/Objectives:** Encouraged by the traditional use of *Cotinus coggygria* Scop. (European smoketree) for its anti-inflammatory and antioxidant properties, and considering the limitations of current therapies for recurrent aphthous stomatitis (RAS), we aimed to develop and evaluate a mucoadhesive oral gel containing *C. coggygria* stem bark extract. **Methods:** A thermosensitive gel was formulated using Carbopol^®^ 974P NF and poloxamer 407, enriched with 5% *C. coggygria* extract (CC gel), and characterized for its organoleptic properties, pH, electrical conductivity, and storage stability over six months. Therapeutic efficacy was assessed in a Wistar albino rat model of chemically induced oral ulcers. Animals were divided into three groups: untreated controls (CTRL), rats treated with gel base (GB), and those treated with CC gel over a 10-day period. Healing progression was monitored macroscopically (ulcer size reduction), biochemically (oxidative stress markers in plasma and tissue), and histologically. **Results:** The CC gel demonstrated satisfactory physicochemical stability and mucosal compatibility. Moreover, it significantly accelerated ulcer contraction and achieved complete re-epithelialization by day 6. Biochemical analyses revealed reduced TBARS and increased SOD, CAT, and GSH levels in ulcer tissue, indicating enhanced local antioxidant defense. Histological evaluation confirmed early resolution of inflammation, pronounced fibroblast activity, capillary proliferation, and full epithelial regeneration in the CC group, in contrast to delayed healing and persistent inflammatory infiltration in the GB and CTRL groups. **Conclusions:** These findings indicate that the CC gel has potential as a natural, topical formulation with antioxidant and regenerative properties for RAS, although further studies, including clinical evaluation, are required to confirm its overall efficacy and long-term safety.

## 1. Introduction

The genus *Cotinus*, belonging to the Anacardiaceae family and comprising two species, *C. coggygria* Scop. (European smoketree) and *C. obovatus* Raf. (American smoketree), has been widely used in traditional medicine. *C. coggygria* Scop., in particular, is notable for its broad geographic distribution, extending from Southern Europe and the Mediterranean region to Southern Russia, Moldova, the Caucasus, Turkey, Central China, and the Himalayas [1,2]. The plant of *C. coggygria* represents a source of valuable essential oils and extracts, which have been used in traditional medicine for the treatment of different diseases, such as skin and mucosal lesions, hypertension, urinary diseases, cough, fever, asthma, hemorrhoids, diabetes, liver disease, and cancer [3]. These findings are supported by the fact that smoketree is rich in a diverse range of polyphenolic secondary metabolites, including tannins, various flavonoid subclasses, phenolic acids, and volatile organic compounds. Notably, its flavonoid profile is characterized by a distinct presence of 5-deoxyflavonoids, with significant amounts of sulfuretin, fisetin, and butein [2,3].

Widely used in traditional medicine across the globe, *C. coggygria* has been the subject of extensive research due to its medicinal properties. It serves as a significant source of compounds that offer a variety of health benefits, with many studies highlighting the biological effects of its components [2]. The pharmacological activity of *C. coggygria* was demonstrated in studies using extracts from young shoots, where phenolic- and tannin-rich fractions showed strong anti-inflammatory and cytotoxic effects [4], supporting its relevance for inflammatory conditions such as aphthous stomatitis. This plant species is also widely recognized for its potent antioxidant properties, as consistently demonstrated in earlier research [5,6,7]. Moreover, topical application of an ointment containing 5% ethanol extract has demonstrated significant potential to accelerate wound healing, particularly in diabetic wounds in rats [8].

Recurrent aphthous stomatitis (RAS) is one of the most common oral ulcerative diseases, characterized by painful, round ulcers with an erythematous border and a white-yellowish pseudomembranous center, primarily affecting the non-keratinized oral mucous membranes [9]. Currently available local therapies for RAS are often inadequate and given the significant impact of this condition on patients’ quality of life, there is an urgent need to develop novel treatments that accelerate oral ulcer healing while minimizing side effects compared to conventional synthetic drugs [10]. While RAS may arise due to systemic conditions or physical trauma, recent research has identified a broader range of potential contributing factors [11]. These include deficiencies in certain vitamins, disruptions in the oral microbiome, blood-related abnormalities, psychological stress, genetic variations, and oxidative stress resulting from an imbalance between oxidants and antioxidants. Oral infections and inflammation are often accompanied by a pathological increase in reactive oxygen species (ROS), leading to oxidative stress, which has been shown to damage essential cellular components such as lipids, proteins, and nucleic acids [12]. This damage can further intensify inflammation and contribute to progressive tissue injury through various molecular pathways. Maintaining ROS balance has emerged as a key therapeutic strategy in the treatment of oral infections and inflammatory disorders. Considering the significant anti-inflammatory, antioxidant, and wound-healing properties of *Cotinus coggygria*, we hypothesized that a formulation containing *C. coggygria* extract could effectively promote healing in RAS.

In order to enable prolonged contact of the *C. coggygria* extract with the oral mucosa, we decided to develop an oral hydrogel, i.e., a mucoadhesive gel with thermoreversible properties. Poloxamer-based hydrogels, especially P407, are favored as thermosensitive gels, because they are colorless, easily washable, non-irritating to skin and mucosa, and exhibit a thermosensitive sol–gel transition: they remain liquid at room temperature (~20–25 °C) and gel near physiological temperature (~30–35 °C), enabling convenient in situ gelation (e.g., via buccal sprays). Because poloxamers alone show weak mucoadhesion in biological fluids, different mucoadhesive polymers are added. E.g., anionic carbomers (polyacrylic acid derivatives) are added to form strong hydrogen bonds with mucin oligosaccharides, thereby enhancing the adhesive strength, residence time, and mechanical robustness of the gel at the application site [13]. Indeed, the recent literature highlights the use of poloxamers with carbomers as complementary gelling/mucoadhesive systems that improve local drug retention and biopharmaceutics for oral delivery. Namely, a recent study showed that thermosensitive–mucoadhesive blends of poloxamer 407 (P407) with Carbopol^®^ (e.g., Carbopol^®^ 934/934P) can gel near body temperature (≈35–37 °C), increase residence time on oral mucosa, and support controlled release—e.g., an in situ gel aimed against mouth ulcer with 14–16% *w*/*v* P407 plus Carbopol^®^ 934P (with choline salicylate/borax) tuned gelation and mucoadhesion [14].

Further, a quercetin in situ gel containing 20.84% *w*/*v* P407 and 0.5% *w*/*v* Carbopol^®^ 934P showed suitable gelation (36 °C), acceptable injectability/viscosity, favorable cell viability, and significant antibacterial activity—underscoring the suitability of P407/Carbopol^®^ matrices for oral cavities [15].

In addition, Li et al. formulated a sprayable Pluronic^®^ (F127/F68) thermogelling system whose mucoadhesion was boosted by carbomer (Carbopol^®^ or Noveon^®^), for localized pain relief in oral mucositis [16]. Díaz-Salmerón et al. developed a P407 (15–17 *w*/*w* %) hydrogel for buccal delivery of dexamethasone (threaded into hydroxypropylbeta-cyclodextrin (HP-β-CD) molecules), where one optimized variant of the hydrogel incorporated Carbopol C971P as the mucoadhesive enhancer [13].

Taking the aforementioned into consideration, the aim of our research was to formulate and characterize a novel oral mucoadhesive gel based on *C. coggygria* extract and to evaluate for the first time its therapeutic effects on aphthous stomatitis in rats through biochemical, macroscopic, and histological analyses. By investigating the potential of *C. coggygria* extract delivered via this mucoadhesive gel, the study aims to offer a natural, targeted therapy capable of enhancing healing outcomes, improving patient quality of life, and providing a safer alternative to conventional synthetic drugs with fewer side effects.

## 2. Materials and Methods

### 2.1. Materials

Poloxamer 407 (Kolliphor^®^ P 407) was obtained from BASF (Ludwigshafen, Germany), while carbomer (Carbopol^®^ 974P NF) was obtained from Lubrizol (Farmsum, The Netherlands), both as gifts. Methanol was purchased from Centrohem (Belgrade, Serbia), and triethanolamine (TEA), acetonitrile and orthophosphoric acid from Sigma Aldrich (St. Louis, MO, USA). *Cotinus coggygria* Scop. (*Rhus cotinus* L.) Anacardiaceae (smoke tree) was collected in Bosnia and Herzegovina. All chemicals for determination of total phenolic content (TPC), total flavonoid content (TFC), and 2,2-diphenyl-1-picryl-hydrazyl-hydrate (DPPH) scavenging capacity, as well as for the in vivo healing efficacy experiments, were purchased from Sigma-Aldrich (St. Louis, MO, USA).

### 2.2. Preparation of C. coggygria Extract

The extraction of bioactive compounds from the plant material was performed using methanol. Methanolic extraction was used because methanol efficiently extracts a wide range of polar and semi-polar phytochemicals. A total of 35 g of finely ground *C. coggygria* bark was extracted with 250 mL of methanol. The mixture was heated in a water bath under reflux at 65 °C for two hours. After the extraction, the mixture was filtered to separate the solid residue from the liquid extract. The same extraction procedure was repeated with the remaining plant material using fresh methanol. The resulting liquid extracts were combined and concentrated to dryness using a rotary vacuum evaporator to ensure complete removal of methanol before in vivo use. The obtained crude extract was stored for further analysis.

### 2.3. HPLC Analysis of the Extract

High-performance liquid chromatography (HPLC) analysis was performed using an Agilent 1260 RR system (Agilent Technologies, Waldbronn, Germany) equipped with a diode-array detector (DAD) capable of scanning across the 190–550 nm range, according to the previously published procedure [17]. The separation of compounds was carried out on a reversed-phase Zorbax SB-C18 column (150 mm × 4.6 mm i.d., 5 μm particle size, Agilent). The mobile phases consisted of 1% (*v*/*v*) orthophosphoric acid in water (solvent A) and acetonitrile (solvent B). A gradient elution program was applied, starting with 90% A from 0 to 2.6 min, gradually decreasing to 85% by 8 min, maintained until 10.8 min. The composition was further reduced to 80% A by 18 min, held constant until 23 min, then decreased stepwise to 70% at 25 min, 50% at 27 min, 20% at 29 min, and 10% at 31 min, followed by a final decrease to 0% A at 34 min, which was held until 35 min. The flow rate throughout the analysis was 1.0 mL/min. Each sample injection was 8 μL, and the column temperature was kept at 40 °C. Detection was conducted at four specific wavelengths: 260, 280, 320, and 360 nm. Compounds were identified by comparing their retention times and UV spectra to those of authenticated standards. Quantification was based on calibration curves generated from the standards, and the results were expressed either as micrograms per gram of dry weight (μg/g dw) for solid samples or milligrams per milliliter (mg/mL) for liquid samples.

### 2.4. Determination of Total Phenolic Content (TPC)

The total phenolic content (TPC) of the methanol extract of *Cotinus coggygria* bark was determined using the Folin–Ciocalteu (FC) colorimetric method, with minor modifications to the original protocol [18]. Gallic acid was used as the reference standard, with a calibration curve constructed using standard solutions in the range of 25–1000 μg/mL. Briefly, 50 μL of the test solution or standard was mixed with 150 μL of distilled water and 1 mL of FC reagent. After 5 min, 800 μL of 7.5% sodium carbonate solution was added. The mixture was incubated in the dark at room temperature for 60 min with occasional shaking, and absorbance was measured at 760 nm. The results were expressed as mg of gallic acid equivalents per gram of dry extract (mg GAE/g DE), based on the calibration curve (y = 0.0026x + 0.0819; R^2^ = 0.9989) [19].

### 2.5. Determination of Total Flavonoid Content (TFC)

The total flavonoid content (TFC) was evaluated by the aluminum chloride colorimetric method, which relies on complex formation between flavonoids and AlCl_3_ [20]. A total of 1 mL of the extract or standard (concentration range: 31.25–1000 μg/mL) was mixed with 200 μL of 10% AlCl_3_ in methanol, 200 μL of 1 M potassium acetate, and 5.6 mL of distilled water. After incubation for 30 min at room temperature, the absorbance was measured at 415 nm. Two sets of calibration curves were used: one with quercetin (y = 0.0020x + 0.0568; R^2^ = 0.9975) and another with rutin (y = 0.0013x + 0.0268; R^2^ = 0.9981). The results were expressed as mg quercetin equivalents (mg QE/g DE) and mg rutin equivalents (mg RE/g DE) [21].

### 2.6. DPPH Radical Scavenging Assay

The DPPH radical scavenging activity of the extract was assessed following a modified protocol based on the reduction of DPPH radicals [22]. A 0.05 mg/mL DPPH solution in methanol was freshly prepared and kept in a dark container until use. Methanolic solutions of the extract and standards (ascorbic acid and Trolox) were prepared in a concentration range of 31.25–1000 μg/mL. A volume of 200 μL of each solution was mixed with 2 mL of DPPH solution. The mixtures were incubated in the dark at room temperature for 30 min, and the absorbance was recorded at 517 nm. The percentage of DPPH radical inhibition was calculated using the following equation:% inhibition = 100 × (A_k_ − A_u_)/A_k_
where A_k_ is the absorbance of the control and A_u_ is the absorbance of the test sample. The IC_50_ values were obtained from nonlinear regression analysis and represent the concentration required to scavenge 50% of the DPPH radical [23].

### 2.7. Preparation of C. coggygria Extract-Loaded Oral Mucoadhesive Gel (CC Gel)

Carbomer (Carbopol^®^ 974P NF) was dispersed in purified water by mechanical stirring (10 min, 500 o/min, Velp Scientifica LS, Pudong New Area, Shanghai, China) in concentration 0.15% *w*/*w*. Afterwards, Poloxamer (Kolliphor^®^ P 407) was added to the carbomer dispersion at a concentration of 20.0%, *w*/*w*. This mixture was kept overnight (12 h) to ensure complete hydration of the gelling agents. The next day, the dispersion was stirred by a mixer (10 min, 500 o/min, Velp Scientifica LS) to ensure complete mixing of the two polymers, and the obtained gel was mixed with 5% *w*/*w* of the dry plant extract. Finally, a 10% *w*/*w* solution of TEA was added for neutralization. The prepared CC gel was stored in the refrigerator at 4 °C until being used. The gel exhibits thermoreversible behavior, remaining in a liquid state at refrigeration temperature (4 °C) and undergoing a sol-to-gel transition to a semi-solid state at room temperature.

A hydrogel without smoketree extract (placebo hydrogel, GB) was prepared using the same procedure as the hydrogel containing the extract, except that no extract was added; instead, purified water was used to make it up to 100%.

### 2.8. Physicochemical Characterization of CC Gel

The long-term stability of the formulated gel was assessed by monitoring its organoleptic characteristics including color, odor, and consistency, alongside pH and electrical conductivity, both 24 h after preparation and after 180 days of storage. The gel was stored in tightly sealed plastic containers under ambient storage at 25 ± 2 °C. In addition, accelerated stability was evaluated using a centrifugation test to assess potential phase separation or physical instability and using rheological measurements after the preparation of the gels and after the thermal stress test [24].

#### 2.8.1. Determination of Organoleptic Properties

The gel’s organoleptic properties, including color, odor, and consistency, were evaluated 24 h after gel preparation and after 180 days of storage at room temperature (22° ± 2 °C) in a well-closed plastic box. Odor was assessed by applying a thin layer of the formulation onto a glass plate and evaluating the scent upon exposure to air. Color was evaluated by spreading a small amount of the sample onto white paper to enhance contrast, and then visually comparing it with the corresponding gel and cream base formulations [25].

#### 2.8.2. Determination of the pH Values

The pH values of the examined formulation were determined by using a digital pH meter (Mettler Toledo, Columbus, OH, USA) calibrated using a standard buffer solution at 22 ± 2 °C. All measurements were repeated three times [24,25].

#### 2.8.3. Determination of the Electrical Conductivity

Electrical conductivity (σ) of the examined formulations was determined by using a conductivity meter (Eutech CON 700, Thermo Fisher Scientific, Shanghai, China) at 22 ± 2 °C. All measurements were repeated three times [24,25].

#### 2.8.4. Assessment of Long-Term Stability of the CC Gel

The long-term stability of the gel containing *Cotinus coggygria* extract was assessed by determination of organoleptic characteristics and alterations in pH and electrical conductivity values during the storage period. The samples were stored for 6 months. The sampling was conducted after 7, 90, and 180 days of the gel storage [25].

#### 2.8.5. Centrifugation Test

The gel samples were centrifuged twice at 3000 rpm for 15 min by using a laboratory centrifuge (Hettich Mikro 120, Hettich, Ste 136L Beverly, MA, USA). Centrifugation was performed at room temperature (22° ± 2 °C) for 24 h after the preparation of the gel. Each sample was subjected to visual inspection in order to detect any changes such as phase separation [24,25].

#### 2.8.6. Rheological Characterization

Rheological characterization of the hydrogels—both those containing the *Cotinus coggygria* extract and the placebo—was performed using steady-shear and oscillatory tests on a Discovery Hybrid Rheometer HR-2 (TA Instruments, New Castle, DE, USA) at 25 °C, with the temperature controlled by a Peltier system. A parallel-plate geometry (25 mm diameter) was employed with a fixed gap of 1000 μm for all measurements. Steady-shear testing comprised three consecutive steps: an ascending shear-rate ramp from 0.1 to 200 s^−1^, a 60 s hold at 200 s^−1^, and a descending ramp from 200 to 0.1 s^−1^. Oscillatory measurements were conducted at 1% strain over an angular frequency range of 0.1–100 rad s^−1^. Samples were analyzed 48 h after preparation and again following a thermal stress test to evaluate physical stability. The thermal stress test was performed by five alternating storage cycles at 4 °C (refrigerator), 20 ± 2 °C (room temperature), and 45 °C (thermostat), with 24 h at each temperature. This thermal stress protocol constitutes an accelerated aging assay often used to predict the long-term stability of gel formulations [26]. The obtained results are presented graphically.

### 2.9. Oral Ulcer Healing Examinations in Wistar Albino Rats

#### 2.9.1. Animals

Forty-five Wistar albino rats (200 ± 20 g) were included in the in vivo examination of oral ulcer healing potential of the novel mucoadhesive gel with *C. coggygria extract* (CC gel). Sample size estimation was conducted using the G*Power 3 software, guided by previously published studies employing a similar methodology [8]. The calculation was based on a significance level of α = 0.05 and a statistical power of 0.80 for a two-tailed independent samples *t*-test comparing groups. Animals were obtained from the Military Medical Academy, Belgrade, Serbia, and housed at a temperature of 22 ± 2 °C, with 12 h of automatic illumination daily. They consumed commercial rat food (20% protein rat food; Veterinary Institute Subotica, Subotica, Serbia) ad libitum. This research was conducted at the Center of Excellence for Redox Balance Research in Cardiovascular and Metabolic Disorders. All procedures were performed according to the EU Directive for the welfare of laboratory animals (2010/63/EU) and the principles of Good Laboratory Practice (GLP).

#### 2.9.2. Oral Ulcer Induction in Rats

Oral mucosal ulcer was induced by exposing rats’ buccal surfaces to glacial acetic acid according to a previously well-established model. Before ulcer induction, rats were anesthetized with a mixture of ketamine (5 mg/kg) and xylazine (10 mg/kg) intraperitoneally. After two days, chronic ulceration was developed, characterized by well-defined borders, and that time point was marked as day 0 [10].

#### 2.9.3. Experimental Design

After confirmation of RAS, the rats were randomly assigned to three groups using pre-generated random number tables to ensure unbiased allocation (15 rats per group):

CTRL group—the ulcer was left without any treatment;

GB group—the ulcer was treated with gel base once daily;

CC group—the ulcer was treated with the 5% *C. coggygria* extract oral mucoadhesive gel once daily.

The examined mucoadhesive gel was topically applied using cotton swab on a plastic stick (amount was 0.5 g) once daily until recovery. The animals were sacrificed at different time points during the 10-day protocol. Five animals from each group were sacrificed after a short-term ketamine/xylazine anesthesia on days 3, 6, and 10. Blood samples were taken for determination of systemic redox status, while buccal tissue samples were collected for determination of local redox status. The researchers who performed group allocation and treatment administration did not conduct outcome assessments or data analysis. These evaluations were carried out independently by separate, blinded researchers to ensure unbiased assessment of the experimental results.

Animals were monitored daily for signs of pain, distress, or behavioral changes, including locomotor activity, posture, piloerection, and food and water intake. Body weight was recorded on day 3, 6, and 10 as an indicator of overall health and systemic toxicity. Any adverse effects at the oral ulcer site or evidence of systemic distress would have prompted immediate intervention. However, all animals tolerated the procedures well, with no signs of excessive pain, infection, or treatment-related complications.

#### 2.9.4. Assessment of the Oral Buccal Ulcer Healing Potential of CC Gel

The establishment and the oral ulcer healing process were photographed and monitored every second day, starting from day 0 until complete healing. The ulcer area was quantified using the ImageJ software (v.1.53; NIH, Bethesda, MD, USA). All images were calibrated prior to analysis using a millimeter scale included in each photograph to ensure measurement accuracy. To minimize bias, images were coded and analyzed in a blinded manner, without knowledge of group allocation. The ulcer margins were manually delineated, and the ulcerated area was calculated in mm^2^. To ensure measurement consistency, two experienced investigators independently performed the analysis, and their results were in close agreement. The degree of healing was expressed as a percentage of contraction of the ulcer area and it was calculated for each rat using the following formula:Ulcer healing rate (%) = (A_0_ − A_t_) × 100/A_0_
where A_0_ and A_t_ represent the initial ulcer area and the ulcer area at the time of observation, respectively.

#### 2.9.5. Biochemical Analysis: Systemic and Local Redox State

Markers of the systemic redox state were determined in blood samples collected from a jugular vein. The concentration of the following pro-oxidants was spectrophotometrically determined in plasma samples: the levels of the superoxide anion radical (O_2_^−^), nitrites (NO_2_^−^), hydrogen peroxide (H_2_O_2_), and the index of lipid peroxidation (measured as thiobarbituric acid reactive substances (TBARS)). Additionally, parameters of the antioxidative defense were determined in the lysate of erythrocytes: activities of superoxide dismutase (SOD) and catalase (CAT) as well as the level of reduced glutathione (GSH) [27].

On the other hand, samples of the aphthous lesions were isolated for determination of local redox status. After isolation, samples were immediately frozen at −80 °C and 0.5 g section of each tissue was homogenized in 5 mL of ice-cold phosphate-buffered saline (pH 7.4). After this procedure, the homogenate was centrifuged at 10,000× *g* at 4 °C for 15 min and the obtained supernatants were used for oxidative stress analysis. The following parameters were spectrophotometrically determined: TBARS, GSH, CAT, and SOD, according to a previously established protocol [10].

#### 2.9.6. Histological Analysis

The isolated aphthous lesion samples were fixed in 4% formalin for 24 h. Tissue sections were then dehydrated in increasing concentrations of alcohol (70–100%), cleaned in xylene, and embedded in paraffin. Sections 5 μm thick were cut and stained with hematoxylin and eosin (H/E) to assess morphological changes. Images of the stained tissues were captured using a digital camera mounted on an Olympus BX51 microscope. The tissue samples were analyzed according to our previously described methodology [10]. Briefly, the following histological scoring protocol was used: (1) Presence of epithelial necrosis, no signs of inflammation; (2) the inflammatory reaction has started, with no new capillary proliferation; (3) the inflammatory reaction is prominent with few capillary proliferations on the basis of the ulcer, but no epithelization at the surface; (4) the inflammatory reaction is decreased, new capillary proliferation has reached the surface, and epithelization has started at the surface; (5) the epithelization is complete.

### 2.10. Statistical Analysis

The Statistical Package for Social Sciences v23.0 (SPSS; IBM Corp., Armonk, NY, USA) was used for the statistical analysis of the obtained results. The Shapiro–Wilk test was used to evaluate the distribution of the data. The parametric tests one-way ANOVA and independent samples *t*-test, and the Kruskal–Wallis H and Mann–Whitney U non-parametric tests, were performed to determine the differences between groups depending on the normal distribution of the data. All results are presented as mean values ± standard deviation (SD). A *p* < 0.05 was considered to be significantly different.

## 3. Results and Discussion

### 3.1. Phytochemical Analysis of the Extract

Numerous phytochemicals have been identified in various parts of the smoketree plant. Research indicates that flavonoids represent the predominant and most biologically significant class of compounds in Cotinus species, with phenolic acids also present in notable amounts [2,17]. The phytochemical profiling of the extract revealed the presence of several phenolic compounds and flavonoids, with sulfuretin being the most abundant (154.62 μg/g dw), followed by fisetin (20.79 μg/g dw) (Table 1). The corresponding chromatogram is provided in the Appendix A. Quercetin and its derivatives, including hyperoside, isoquercitrin, and trace amounts of quercitrin, were present in the extract, indicating a diverse flavonoid profile with potential antioxidant and anti-inflammatory activity [28]. Phenolic acids such as gallic, chlorogenic, and ellagic acid were detected in the extract in low concentrations, contributing to its overall antioxidant potential [29].

### 3.2. Antioxidant Activity of Cotinus coggygria Bark Extract

The results presented in Table 2 show that the methanol extract of *C. coggygria* bark exhibited a remarkably high total phenolic content (TPC: 1504.00 ± 7.15 mg GAE/g DE), as well as substantial levels of total flavonoids, expressed as both quercetin (TFC Q: 875.87 ± 2.10 mg QE/g DE) and rutin equivalents (TFC R: 1460.79 ± 3.44 mg RE/g DE). Table 3 demonstrates the DPPH radical scavenging activity for this extract (IC_50_: 292.02 ± 24.28 µg/mL), which, while significant, is considerably less potent than the reference antioxidants ascorbic acid (IC_50_: 9.12 ± 1.91 µg/mL) and Trolox (IC_50_: 14.22 ± 3.92 µg/mL).

Compared to recent studies, the phenolic and flavonoid contents reported herein are notably higher than those reported for similar extracts or related fractions. For instance, a previous study revealed TPC values in ethyl acetate fractions of *C. coggygria* up to 929.8 mg/g, with a flavonoid content around 35.5 mg/g and a much lower DPPH IC_50_ (1.7 µg/mL), indicating higher antioxidant efficacy for that particular fraction but a substantially lower phenolic yield compared to our methanol bark extract. Shaboyan et al. also reported a lower TPC and TFC in leaf and branch extracts from Armenian flora, with DPPH IC_50_ values between 64 and 140 µg/mL, still lower than for the present bark extract, but overall the TPC and TFC values were lower [30]. Recent detailed compositional analyses further confirmed the abundance of numerous flavonoid and phenolic constituents in *C. coggygria*, though without reporting TPC or DPPH values matching those found in this work [17]. Although the methanol bark extract showed relatively weak radical scavenging activity in the DPPH assay (IC_50_ ≈ 292 µg/mL) compared to the pure reference antioxidants (ascorbic acid and Trolox), this finding is consistent with the complex nature of crude plant extracts, where active constituents are present alongside non-active matrix compounds. Thus, the DPPH result should not be considered a direct predictor of the in vivo efficacy.

### 3.3. Physicochemical Characterization of the CC Gel

Drawing on the traditional use of *C. coggygria* in herbal medicine and its well-documented bioactive properties, this study focused on creating a mucoadhesive gel tailored for oral use in treating aphthous ulcers. In addition to selecting an effective natural extract, the formulation strategy emphasized the importance of a delivery system capable of maintaining prolonged and close contact with the oral mucosa for optimal therapeutic effect [10]. The results presented below include an evaluation of the gel’s physicochemical stability during storage, followed by an in vivo assessment of its safety and therapeutic potential in a rat model of aphthous stomatitis.

#### 3.3.1. Organoleptic Characteristics and Physical Appearance of the CC Gel

The formulated *C. coggygria* (CC) mucoadhesive gel was first evaluated for its organoleptic properties, as these characteristics are essential indicators of product uniformity, patient acceptability, and early physical stability [31]. The initial gel base was white, homogeneous, and odorless. Upon incorporation of *C. coggygria* extract, the gel adopted a brown color and developed a pleasant scent characteristic of the extract (Figure 1). This thermosensitive poloxamer 407 system undergoes a sol–gel transition near physiological temperature, enabling easy application as a liquid and in situ gelation on the oral mucosa—thereby increasing viscosity and supporting sustained release of bioactive constituents [13,14]. In parallel, carbomer imparts strong mucoadhesion (via hydrogen bonding to mucins), prolonging residence and limiting salivary wash-out; comparable P407/carbomer matrices likewise show improved retention and favorable performance in oromucosal models [13,15,16].

The gel samples exhibited no significant changes in organoleptic properties over the 6-month observation period. Both formulations remained homogeneous, with no noticeable alterations in color or odor, and preserved their semi-solid consistency and homogeneity over time, with no signs of phase separation or instability (Table 4). These findings indicate satisfactory physical stability of the gels under ambient storage conditions, supporting their suitability for further application and testing (Table 4).

#### 3.3.2. pH Values and Electrical Conductivity of the CC Gel

The pH and electrical conductivity of both the gel base (GB) and the *C. coggygria*-enriched gel (CC) were monitored over a period of 180 days to evaluate their physicochemical stability under ambient storage conditions (22 ± 2 °C) (Figure 2A,B). As shown in Figure 2A, the CC gel consistently exhibited a slightly higher pH than the gel base, likely due to the inherent properties of the *C. coggygria* extract. Over the six-month storage period, the pH of both formulations showed slight decreases. The GB formulation exhibited a pH of 6.50 ± 0.01 at day 0, which decreased to 6.43 ± 0.01 at day 60 and further to 6.11 ± 0.02 at day 180. Similarly, the CC formulation started with a pH of 6.84 ± 0.01 on day 0, which slightly decreased to 6.80 ± 0.02 at 60 days and reached 6.52 ± 0.01 at 180 days. Although slight reductions in pH were observed in both formulations over time, these variations remained within the physiologically acceptable range for oral mucosal products (5.5–7.9), indicating that the formulations maintained their chemical stability and mucosal compatibility throughout the storage period [32,33]. Monitoring pH is a critical aspect of evaluating formulation stability, as significant deviations can indicate degradation or incompatibility among components [32]. However, in this case, the changes observed were not substantial and are unlikely to affect the product’s safety or efficacy. Furthermore, the combination of plant-based actives and the used polymers did not result in any notable destabilization, suggesting the CC gel is both stable and suitable for oral application.

Similarly, electrical conductivity, which indicates the level and movement of ions within a formulation, offers important insights into the physicochemical stability of gels during storage. Changes in conductivity correspond to variations in ion transport, swelling dynamics, and the overall integrity of the gel’s structure [34]. In this study, the gel base–placebo gel (GB) initially showed an electrical conductivity of 80.2 ± 1.1 µS/cm, which gradually decreased to 71.1 ± 3.31 µS/cm after 180 days of storage. Similarly, the *C. coggygria* (CC) gel exhibited an initial conductivity of 77.6 ± 1.88 µS/cm, declining to 68.7 ± 2.1 µS/cm over the same period (Figure 2B). The gradual reduction in electrical conductivity observed in both formulations likely reflects subtle changes that may be associated with slight modifications in the internal polymeric network, such as alterations in hydrogen bonding and crosslinking density, which are known to influence key functional properties, including gelation behavior, mucoadhesiveness, and mechanical stability [35]. Despite these alterations, the conductivity values remained relatively stable and consistent with the slight pH decline, suggesting the formulations maintained their structural integrity, physicochemical stability, and ionic balance throughout the observation period. Together with the pH data where both gels retained values within the physiologically acceptable range, these findings suggest that the CC gel formulation remains physically and chemically stable under ambient storage conditions for at least six months, supporting its suitability for oral mucosal application.

#### 3.3.3. Physical Stability Determined by the Centrifugation Test

Centrifugation is widely used as a stress test to evaluate the physical stability of semi-solid formulations by accelerating potential destabilization processes, such as syneresis, phase separation, or precipitation [36]. In the case of our gel containing *C. coggygria* extract, no signs of structural breakdown, liquid separation, or sedimentation were observed after centrifugation, indicating that the formulation maintains physical integrity under stress conditions.

#### 3.3.4. Physical Stability Determined by Rheological Measurements

The flow behavior of the investigated hydrogels is presented using flow curves (rheograms) depicting the dependence of shear stress (τ) on the shear rate (D). Flow curves for the gel samples—i.e., the hydrogels containing *C.coggygria* extract and the gel base—recorded 48 h after preparation and after five thermal-stress cycles are shown in Figure 3.

Forty-eight hours after preparation, the hydrogels with and without the extract exhibited non-Newtonian flow, as viscosity was not constant but depended on shear rate, and plastic behavior characterized by a yield stress (i.e., a minimum shear stress below which the system resists deformation and above which flow begins) (Figure 3). Both samples followed a Herschel–Bulkley flow model (R^2^ > 0.99), determined from the ascending flow curves using the rheometer software (version 5.1, TRIOS, TA Instruments, New Castle, DE, USA). Such plastic flow is desirable for semi-solid preparations, and higher yield-stress values indicate a more organized internal structure. Both gels showed minimal thixotropy, as evidenced by the very small hysteresis areas on the flow curves. The accelerated aging (cyclic temperature storage) caused only slight changes in the flow behavior of the extract-loaded hydrogel, a trend also observed for the placebo; incorporation of the extract did not alter the type of flow.

In addition to the flow curves, the maximum and minimum apparent viscosities were compared for the CC gel before and after accelerated aging. The maximum apparent viscosity (at the lowest shear rate) reflects the structure at rest, whereas the minimum apparent viscosity (at the highest shear rate) reflects structural breakdown under shear. For the CC gel, the maximum apparent viscosity measured at a shear rate of 3.98 s^−1^ was 150.33 Pa·s at 48 h, increasing to 183.72 Pa·s after accelerated aging. The GB gel showed a slight decrease from 150.86 to 144.96 Pa·s. The minimum apparent viscosity of the CC hydrogel (measured at 200 s^−1^) rose from 5.45 to 6.41 Pa·s after aging, while in the GB gel it decreased from 5.90 to 5.53 Pa·s.

A key parameter for semi-solid formulations is the yield stress, which is advantageous for topical application: such systems offer low resistance during spreading (high shear) yet do not flow at rest, preventing dripping from a spatula or fingers until the pressure exceeds the yield stress. Yield stress is a reliable indicator of stability [37,38] and can be used to predict long-term stability [37]. The yield stress of the extract gel increased from 497.91 Pa (48 h) to 631.07 Pa after the stress test while the placebo rose from 460.49 to 532.41 Pa, indicating a marked increase in both cases. These findings point to substantial structuring of the gels during accelerated aging, producing systems that resist external forces to a greater extent before flow begins. High yield-stress values not only suggest improved stability but may also indicate dominant elastic behavior, given reported correlations between yield stress and elastic parameters in some formulations [39].

Gel viscoelasticity was assessed via the storage modulus (G′), loss modulus (G″), and loss tangent (tan δ = G″/G′). The dependence of G′ and G″ on strain was monitored at 1 Hz, and oscillatory parameters were also recorded over 0.015–15 Hz at constant strain. G′, G″, and tan δ are informative for viscoelastic properties and for predicting long-term stability [37]. Both the CC gel and GB gel exhibited dominant elastic behavior (G′ > G″) (Figure 4). At 1 Hz, the extract gel showed G′ = 13,457.9 Pa and G″ = 1131.27 Pa, while the placebo showed G′ = 13,995.3 Pa and G″ = 1712.11 Pa, clearly indicating elastic dominance. The G′ curve did not intersect the G″ curve during measurement (Figure 4), further supporting good structuring or “strength”. Additionally, tan δ < 0.5 (0.08 for the extract gel and 0.12 for the placebo) and tan δ values below 1 indicate prevailing elastic properties and a highly interconnected gel network [38,40].

During accelerated aging of the CC gel (Figure 5A, the viscoelastic behavior changed as a result of further network structuring. The storage modulus (G′) increased to 18,849.5 Pa, the loss modulus (G″) to 1330.28 Pa, and the loss tangent (tan δ) was 0.7, indicating a slight additional increase in the elastic contribution during the stress test, i.e., formation of new bonds between polymer particles. The stress test did not reduce the stability of the CC hydrogel containing the smoketree extract.

During accelerated aging of the GB gel (Figure 5B), only minor changes in viscoelastic behavior were observed. Post-stress-test values of the storage modulus differed only slightly from those measured 48 h after preparation (Figure 5B), while the loss modulus varied somewhat more but remained close to its initial value. Specifically, G′ increased marginally to 14,504.7 Pa, whereas G″ decreased to 1539.84 Pa, yielding tan δ = 0.11 (vs. 0.12 at 48 h), indicative of a negligible shift toward greater elastic contribution during the stress protocol. Overall, the stress test did not compromise the stability of the GB gel.

Rheological measurements performed after preparation and following accelerated aging (thermal-stress cycling) showed that both the extract-loaded CC gel and placebo gel GB exhibit high yield stress and dominant elastic over viscous behavior—features consistent with long-term stability. Only minor parameter shifts were observed, attributable to progressive network structuring typical in the days to months after manufacture. In the CC gel, increases in minimum/maximum apparent viscosity, yield stress, and storage modulus (G′) indicate strengthening of the gel network via additional polymer–polymer interactions. Accelerated aging thus supports that incorporating the smoketree extract does not compromise stability; no rheological changes predictive of system instability or unsuitability for use were detected in either gel.

### 3.4. In Vivo Oral Ulcer Healing Potential of CC Gel

The potential of the CC gel formulation to promote the repair of buccal mucosal ulcers was assessed over a 10-day treatment period in vivo in rats. A chemically induced rat model of aphthous stomatitis was employed, as it closely replicates the clinical, inflammatory, and healing features of human oral ulcers. Given the anatomical and physiological similarities between rat and human oral mucosa, this model was particularly suitable for testing the efficacy of locally applied therapeutic agents, including the developed CC gel [10,41,42]. The efficacy of the formulation was assessed through a combination of clinical observation, biochemical analysis, and histological examination.

#### 3.4.1. Body Weight, Food and Water Intake, and Health Status

Animals were observed daily for indications of pain, discomfort, or changes in behavior, including movement, posture, grooming habits, and food and water consumption. Body weight was monitored on days 0, 3, 6, and 9 to assess the overall health and possible systemic effects of the treatments. Monitoring body weight is important, as changes can reflect discomfort, altered food intake, or treatment effects. Although all groups showed relatively similar trends, rats treated with the CC gel exhibited slightly higher food and water intake (Figure 6A,B) and a modest increase in body weight (Figure 6C) compared to the control and gel base groups. In contrast, the control and gel base groups showed only minimal weight gain, likely due to oral ulcer discomfort. These findings suggest that the CC gel may alleviate oral pain and support normal feeding behavior, contributing to overall well-being.

#### 3.4.2. Ulcer Healing Rate

Ulcer surface area was systematically measured in all experimental groups (CC, gel base, and control) every other day starting from day 0, providing a detailed overview of the healing dynamics. As presented in Figure 7 and Figure 8, animals treated with the CC gel showed a notably enhanced rate of ulcer size reduction compared to both the GB and control groups. An accelerated healing effect was already apparent by the second day of treatment, indicating a rapid onset of therapeutic action, and this enhanced recovery continued consistently. Importantly, by day 6, complete epithelial closure (100% contraction) was observed in the CC group, representing nearly a two-fold improvement relative to the other groups. The observed outcomes highlight the significant ulcer-healing efficacy of the *C. coggygria*-based gel and suggest its promise for application in the treatment of oral mucosal injuries.

The enhanced buccal wound healing observed with CC gel in our in vivo model aligns with previous in vitro findings, where *C. coggygria* extract stimulated fibroblast migration and contributed to wound repair processes. These results reinforce the role of CC-derived bioactives in supporting early tissue regeneration, particularly through fibroblast activation and extracellular matrix remodeling [43]. The healing effects may be attributed to sulfuretin, a predominant compound in *C. coggygria* extract, known for its antimicrobial effects and capacity to mitigate tissue inflammation important for mucosal repair [44,45].

One of the main challenges for oral mucosal drug delivery systems is the rapid clearance caused by continuous saliva flow, which limits the time the drug remains at the site of action [46]. The developed mucoadhesive CC gel formulation overcomes this by enhancing adhesion to the mucosal surface, prolonging drug retention and bioavailability, which may explain the observed protective and accelerated healing of the oral buccal tissue.

#### 3.4.3. Oxidative Stress Parameters

Extensive evidence indicates that oxidative stress plays a key role in the development and persistence of RAS, highlighting the importance of therapeutic approaches targeting the reduction of reactive oxygen species (ROS)-induced damage [47]. The intricate balance between oxidants and antioxidants plays a critical role in the pathogenesis of RAS, where excessive ROS overwhelms the body’s defense systems, leading to oxidative damage of cellular components and contributing to the chronic inflammatory state observed in RAS patients [48]. Moreover, numerous data support the fact that oxidative imbalance is commonly associated with various oral mucosal diseases, including RAS, oral lichen planus, and even premalignant lesion [49]. In our study, oxidative stress parameters were monitored in both ulcer tissue and blood samples to evaluate potential systemic and local redox alterations following treatment.

##### Buccal Tissue Redox State

Figure 9 presents the dynamics of oxidative stress markers (TBARS) and antioxidant parameters (SOD, CAT, GSH) in oral ulcer tissue over 3, 6, and 9 days following treatment with CC gel, gel base, or no treatment (control). The TBARS levels were significantly lower in the CC gel-treated group compared to both the gel base and control groups at all three time points (3, 6, and 9 days), indicating attenuation of lipid peroxidation in the ulcer tissue. The antioxidant parameters (SOD, CAT, and GSH) were significantly elevated in the CC gel-treated group, with the highest values recorded on day 3, followed by a gradual decline through days 6 and 9. These temporal changes reflect the dynamic nature of oxidative stress and the healing process during aphthous stomatitis.

The early increase in SOD and CAT suggests an enhanced enzymatic response, i.e., SOD catalyzing the dismutation of O_2_^−^ and CAT decomposing H_2_O_2_, while elevated GSH supports non-enzymatic redox balance [50]. This pattern demonstrates that CC gel promotes a coordinated antioxidant defense, effectively mitigating oxidative stress throughout the different phases of ulcer healing in aphthous stomatitis. Overall, the significant differences between the CC gel and control groups across all markers and time points indicate that the CC gel enhances the tissue’s antioxidant capacity and reduces oxidative damage.

Additionally, the gel base did not significantly differ from the untreated control, highlighting that the antioxidant effects are attributable to the active compounds in the CC gel formulation. Also, the mucoadhesive nature of the gel ensures prolonged contact with the ulcer surface, enhancing local retention and bioavailability of the antioxidant-rich *C. coggygria* extract. This likely contributes to sustained redox modulation at the site of injury and supports buccal tissue regeneration without systemic interference. These observations are in accordance with previous research on cutaneous wound models, where *C. coggygria* extract significantly increased glutathione levels while reducing malondialdehyde content [51]. Sulfuretin has been previously demonstrated to exhibit significant antioxidant properties that may contribute to tissue repair and inflammation reduction [52]. When delivered via the Carbopol^®^ 974P–poloxamer 407 gel matrix, its controlled release likely enhances local antioxidant effects, ultimately supporting enhanced healing of oral ulcerative lesions. Our findings collectively highlight the potential of the CC gel approach for managing oxidative stress-driven tissue damage in aphthous stomatitis.

##### Systemic Redox State

Pro-oxidants such as H_2_O_2_, NO_2_^−^, TBARS, and O_2_^−^ were determined in plasma samples, while antioxidants such as SOD, CAT, and GSH were assessed in erythrocyte samples, with the results presented in Figure 10 and Figure 11, respectively. Our findings demonstrate that treatment with the CC oral gel did not induce significant systemic changes in oxidative stress markers, except for a notable reduction in O_2_^−^ levels and elevation in SOD activity after 10 days of protocol. The observed rise in SOD activity after CC gel application is consistent with the reduced levels of O_2_^−^, since SOD catalyzes the dismutation of O_2_^−^ into H_2_O_2_ [50]. These slight changes in markers of oxidative stress suggest that the antioxidant effects of the formulation were primarily localized at the ulcer site. Such localized activity is particularly beneficial, as it enables targeted therapeutic action at the site of inflammation and tissue injury, promoting efficient mucosal healing while maintaining systemic redox homeostasis [10].

Importantly, the absence of intense systemic oxidative stress alterations may be attributed to the confined nature of the induced lesions, which were superficial and did not involve deep tissue injury or chronic inflammation [49]. The moderate intensity and short duration of the lesions treated with CC likely did not trigger a robust systemic inflammatory response capable of disrupting overall oxidative balance. This aligns with previous understanding that acute, localized mucosal injuries without deep tissue involvement or chronic inflammation tend to elicit oxidative stress primarily at the site of insult [49]. Furthermore, the lack of prominent systemic oxidative changes is an expected outcome considering the formulation’s use of carbomer and poloxamer 407. These mucoadhesive polymers enhance the retention and localized absorption of active compounds at the site of application, thereby increasing local therapeutic efficacy while minimizing systemic exposure [18].

### 3.5. Histological Alterations

A histological score analysis of buccal ulcer tissue collected on days 0, 3, 6, and 9 is summarized in Table 5. The CC gel group exhibited superior histological healing scores compared to the control and gel base groups, which may be attributed to the combined protective and bioactive effects of the CC formulation. According to the scoring system (Table 3), the gel base group demonstrated a faster improvement than untreated rats, likely due to the physical barrier properties of the gel base. Notably, the CC gel group showed the most accelerated healing, reaching the maximum score of 5.00 ± 0.00 by day 6, indicating enhanced epithelial regeneration and connective tissue remodeling. This suggests that the active polyphenolic compounds in the CC gel significantly promote mucosal healing beyond the effects of the gel base alone.

Microscopic examination of the rat buccal mucosa on day 3 revealed epithelial necrosis, acute subepithelial inflammatory infiltrates predominantly composed of neutrophils, partial ulceration, and dilated blood vessels across all groups. By day 6, a moderate chronic inflammatory response, characterized by fibroblast proliferation and organized collagen fibers, was observed in both the gel base and negative control groups. In contrast, the CC-treated group exhibited remodeled connective tissue and complete ulcer resolution. By day 9, all groups, including the control and gel base groups, showed connective tissue formation indicative of healing, with healing scores approaching those of the CC gel group (Figure 12).

The enhanced tissue regeneration observed in the CC-treated group is plausibly linked to the diverse polyphenolic profile of the *C. coggygria* extract, recognized for its multifaceted biological activities. The structural improvements achieved with CC treatment closely paralleled the biochemical findings, namely, reduced TBARS levels and elevated antioxidant defenses, which were most prominent at day 3 and gradually declined over time. Our findings closely align with those from a study on topical ethanol extract of *C. coggygria* in rat cutaneous wounds, which demonstrated significantly improved tissue regeneration [51]. By scavenging reactive oxygen species and modulating inflammatory responses, the polyphenols in CC gel likely preserved mucosal integrity and enhanced re-epithelialization, resulting in superior healing compared to the gel base and control groups. Collectively, these results demonstrate that CC gel supports both morphological restoration and redox balance in the oral mucosa, thereby promoting efficient healing of aphthous ulcers.

Our findings are consistent with a previous study on *C. coggygria* ointment, which reported accelerated wound healing, elevated hydroxyproline levels, and improved antioxidant defense in diabetic excision wounds [8]. Moreover, previous research on herbal oral gels and other plant-based formulations for RAS, including formulations with extracts from aloe vera, *Galium verum*, Sage, *Musa acuminata*, etc., has demonstrated notable improvements in oral ulcer healing through clinical, in vitro, and in vivo studies [10,53,54]. Nevertheless, these findings cannot be directly compared to our *C. coggygria* gel, as the previous studies employed different plant extracts, polymer systems, and study designs, whereas our formulation combines *C. coggygria* extract with a dual-polymer mucoadhesive system in a rat model of aphthous stomatitis.

Although the present study primarily focused on the development of a *C. coggygria* gel for oral ulcer healing, evaluating systemic safety is essential to ensure its long-term applicability and absence of adverse effects. The CC gel contains natural extract and following short-term application to small oral lesions, systemic absorption of extract components is expected to be minimal, making adverse systemic effects unlikely. This is supported by daily monitoring of body weight and clinical observations, including locomotor activity, posture, grooming, and food and water intake, which revealed no signs of local irritation, behavioral changes, or systemic toxicity. Nevertheless, a comprehensive systemic safety evaluation, including serum chemistry and organ function markers, should be performed in future studies to fully confirm the absence of any systemic effects.

Building on the prominent healing, antioxidant, and histological benefits demonstrated in our study, it is important to emphasize the novelty of our CC gel formulation in the context of RAS treatment. The novelty of our study lies in the formulation and evaluation of a unique combination of *C. coggygria* extract with a synergistic polymer matrix of Carbopol^®^ 974P NF and poloxamer 407. This combination not only ensures strong mucoadhesion to the oral mucosa, effectively addressing the critical challenge of rapid saliva wash-out, but also enables sustained and controlled release of bioactive compounds [55]. To our knowledge, this is the first study to use this specific plant extract with this polymer combination, supported by in vivo validation in aphthous stomatitis rat model. This highlights the distinctive contribution compared to previous studies on *C. coggygria* extract, which primarily focused on extract characterization and in vitro activity, or other studies on herbal oral gels. Consequently, the CC gel enhances retention time at the ulcer site, which likely contributes to accelerated tissue regeneration, improved redox balance, and superior histological outcomes compared to the gel base and untreated controls.

There are certain limitations in the study design, such as the absence of a comparison between the CC gel and other commercial products for the treatment of RAS, which would provide a more complete understanding of its relative effectiveness. In addition, only a single concentration (5%) of *C. coggygria* extract was evaluated, and comprehensive safety assessments, including serum chemistry, organ function markers, and systemic toxicity, were not performed. Since the findings are based on a preclinical animal model, future studies should address these limitations to further validate the efficacy and safety of the CC gel in clinical settings.

## 4. Conclusions

This study presents the successful formulation and preclinical validation of a novel mucoadhesive oral gel containing *C. coggygria* extract, designed for the treatment of RAS. The gel exhibited favorable physicochemical stability, ensuring its suitability for application to the oral mucosa. Incorporating the smoketree extract did not compromise the CC gel stability as no rheological changes predictive of system instability or unsuitability for use were seen in the gel. In vivo investigations demonstrated that the CC gel effectively enhanced tissue regeneration, mitigated oxidative damage, and improved redox homeostasis at the site of injury. The observed therapeutic effects are attributed to both the bioactive properties of *C. coggygria* and the sustained local delivery afforded by the mucoadhesive polymer matrix. Overall, the CC gel shows potential as a natural topical approach for RAS management; however, further clinical studies are essential to confirm its efficacy and safety in human subjects.

## Figures and Tables

**Figure 1 pharmaceutics-17-01293-f001:**
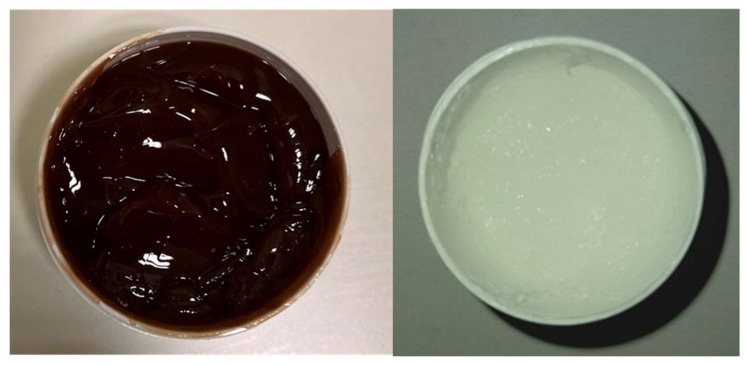
Photographs of the observed formulations 24 h after preparation: gel base (**right**) and gel containing *C. coggygria* extract (**left**).

**Figure 2 pharmaceutics-17-01293-f002:**
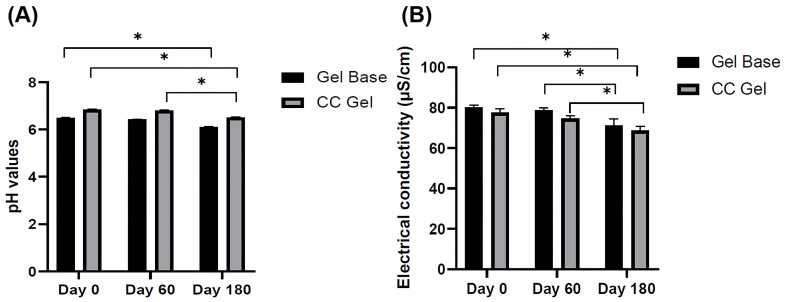
pH values (**A**) and electrical conductivity values (**B**) of GB gel and CC gel after 1, 60, and 180 days of preparation and storage at 22 ± 2 °C. CC gel—gel loaded with *C. coggygria* extract. Measurements were carried out in triplicate and data are presented as mean ± SD (*n* = 3). One-way ANOVA followed by Tukey’s post hoc test was used for statistical analysis. *p* < 0.05 indicates a statistically significant difference compared to day 0, while * *p* < 0.05 indicates a statistically significant difference compared to day 60.

**Figure 3 pharmaceutics-17-01293-f003:**
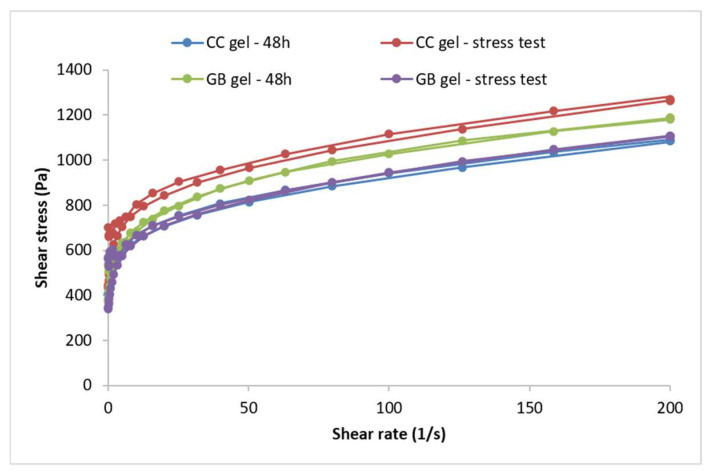
Flow curves of the CC gel and GB gel recorded 48 h after preparation and after the thermal-stress test. CC gel—gel loaded with *C. coggygria* extract, GB—placebo gel.

**Figure 4 pharmaceutics-17-01293-f004:**
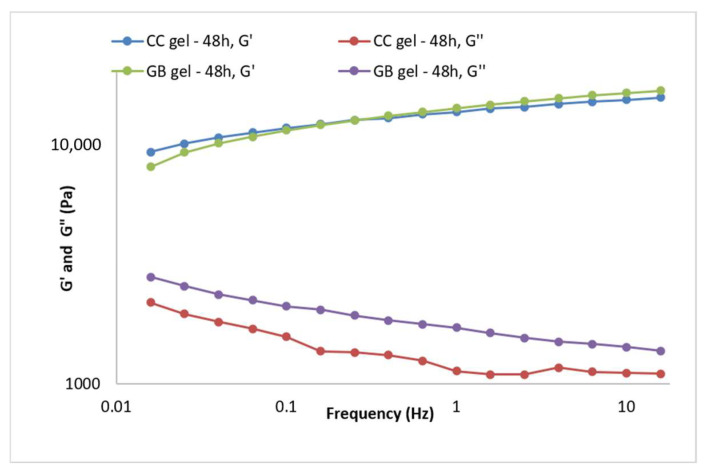
Frequency sweep of the CC gel and GB (placebo) gel 48 h after preparation.

**Figure 5 pharmaceutics-17-01293-f005:**
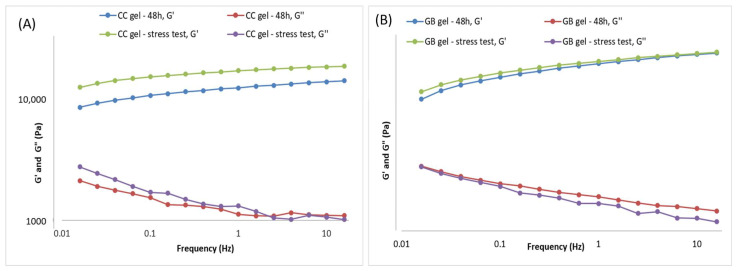
Frequency sweep of the gels 48 h after gel preparation and after the thermal-stress test of the gels: (**A**) CC gel and (**B**) GB (placebo) gel.

**Figure 6 pharmaceutics-17-01293-f006:**
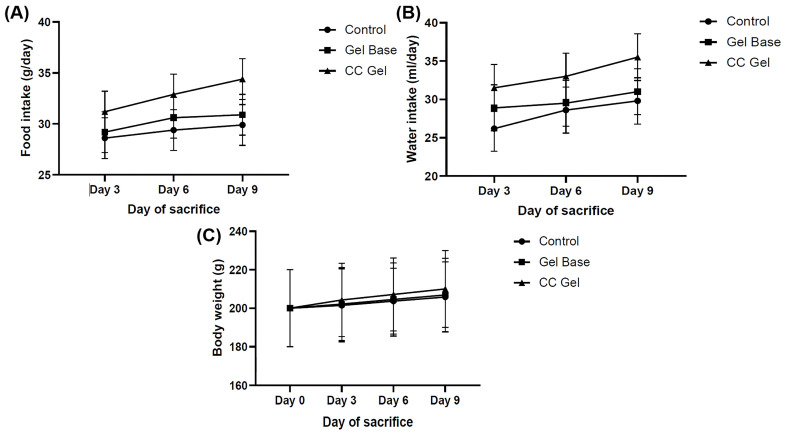
Impact of CC gel treatment on food intake (**A**), water consumption (**B**) and body weight. (**C**). CC gel—ulcers treated with 5% *C. coggygria* extract gel; Gel base—ulcers treated with gel base; Control—untreated ulcers. Data represent mean values from five animals per group (*n* = 5). Statistical analysis was performed using one-way ANOVA followed by Tukey’s post hoc test.

**Figure 7 pharmaceutics-17-01293-f007:**
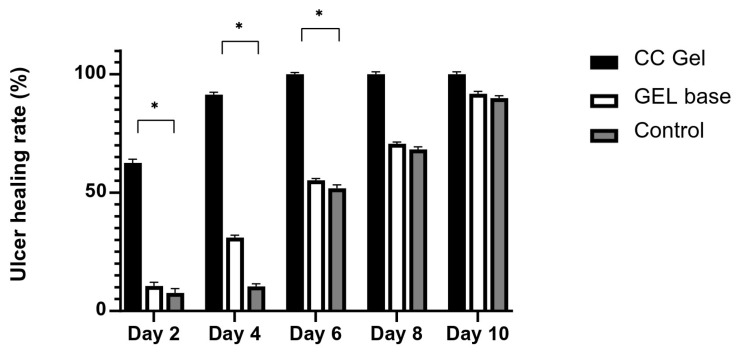
Impact of CC gel treatment on the healing progression of buccal ulcers. CC gel—ulcers treated with 5% *C. coggygria* extract gel; Gel base—ulcers treated with gel base; Control—untreated ulcers. Data represent mean values from five animals per group (*n* = 5). Statistical analysis was performed using one-way ANOVA followed by Tukey’s post hoc test. * *p* < 0.05 indicates statistically significant differences between groups.

**Figure 8 pharmaceutics-17-01293-f008:**
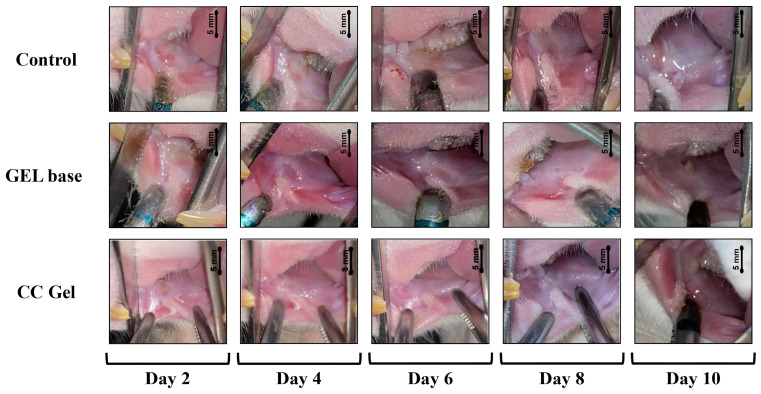
Representative images of oral ulcer progression in rats following various treatments. CC gel—ulcers treated with 5% *C. coggygria* extract gel; Gel base—ulcers treated with gel base; Control—untreated ulcers.

**Figure 9 pharmaceutics-17-01293-f009:**
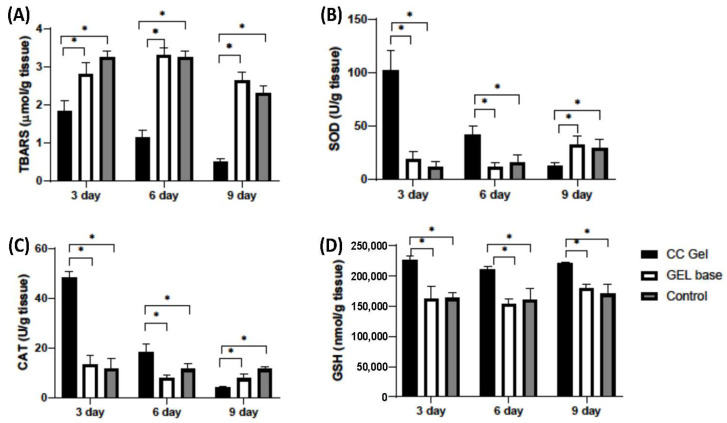
Impact of CC gel treatment on the oxidative stress markers in ulcer tissue samples. CC gel—ulcers treated with 5% *C. coggygria* extract gel; Gel base—ulcers treated with gel base; Control—untreated ulcers. (**A**) TBARS; (**B**) SOD; (**C**) CAT; (**D**) GSH. Data represent mean values from five animals per group (*n* = 5). Statistical analysis was performed using one-way ANOVA followed by Tukey’s post hoc test. * *p* < 0.05 indicates statistically significant differences between groups.

**Figure 10 pharmaceutics-17-01293-f010:**
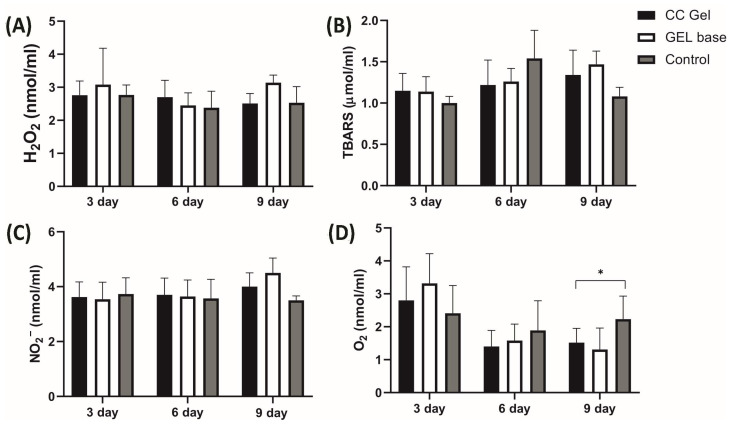
Impact of CC gel treatment on the oxidative stress markers in plasma samples. CC gel—ulcers treated with 5% *C. coggygria* extract gel; Gel base—ulcers treated with gel base; Control—untreated ulcers. Data represent mean values from five animals per group (*n* = 5). (**A**) H_2_O_2_; (**B**) TBARS; (**C**) NO_2_^−^; (**D**) O_2_^−^. Statistical analysis was performed using one-way ANOVA followed by Tukey’s post hoc test. * *p* < 0.05 indicates statistically significant differences between groups.

**Figure 11 pharmaceutics-17-01293-f011:**
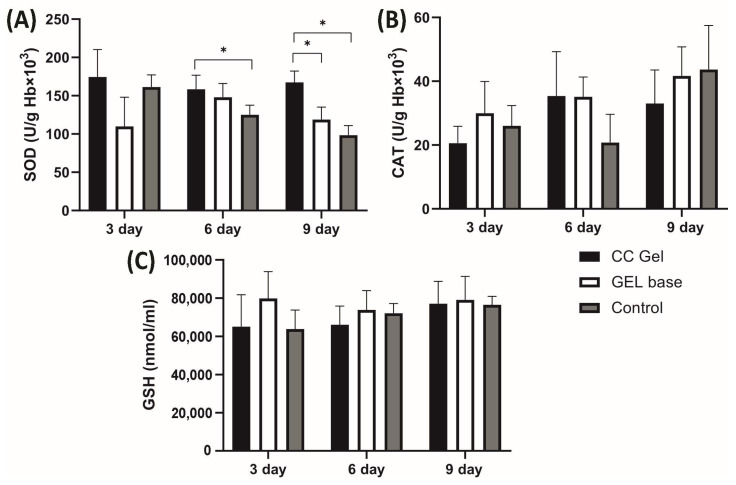
Impact of CC gel treatment on the antioxidant markers in erythrocyte samples. CC gel—ulcers treated with 5% *C. coggygria* extract gel; Gel base—ulcers treated with gel base; Control—untreated ulcers. Data represent mean values from five animals per group (*n* = 5). (**A**) SOD; (**B**) CAT; (**C**) GSH. Statistical analysis was performed using one-way ANOVA followed by Tukey’s post hoc test. * *p* < 0.05 indicates statistically significant differences between groups.

**Figure 12 pharmaceutics-17-01293-f012:**
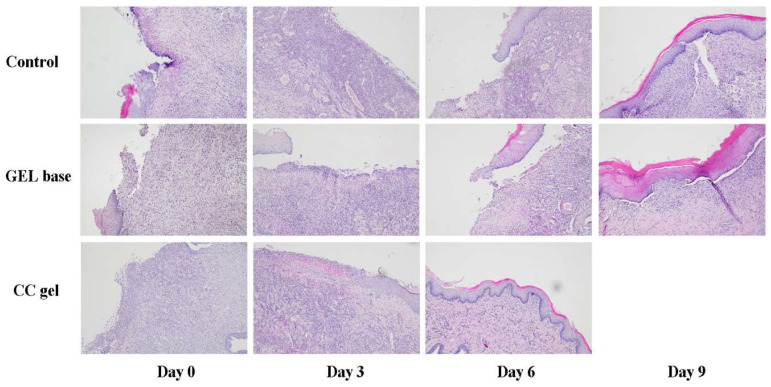
Microphotographs of H/E staining of the oral ulcer on days 0, 3, 6, and 10 (magnification 100×).

**Table 1 pharmaceutics-17-01293-t001:** Content (μg/g dw) of polyphenolic compounds in *C. coggygria* methanolic bark extract.

Polyphenolic Compounds	Content (μg/g dw)
Fisetin	20.79 ± 1.03
Hyperoside	5.62 ± 0.18
Rutin	0.84 ± 0.03
Isoquercitrin	2.11 ± 0.11
Gallic acid	1.17 ± 0.04
Chlorogenic acid	0.90 ± 0.02
Ellagic acid	0.57 ± 0.02
Quercetin	1.01 ± 0.05
Quercitrin	tr. *
Sulfuretin	154.62 ± 7.63

* tr.—traces.

**Table 2 pharmaceutics-17-01293-t002:** Total phenolic content and total flavonoid content of investigated *Cotinus coggygria* bark methanol extract.

	TPC (mg GAE/g DE)	TFC Q (mg QE/g DE)	TFC R (mg RE/g DE)
CC	1504.00 ± 7.15	875.87 ± 2.10	1460.79 ± 3.44

CC—*Cotinus coggygria* bark methanol extract; TPC—total phenolic content; TFC—total flavonoid content; GAE—gallic acid equivalent; QE—quercetin equivalent; RE—rutin equivalent; DE—dry extract.

**Table 3 pharmaceutics-17-01293-t003:** DPPH radical scavenging activity of *Cotinus coggygria* bark methanol extract and reference antioxidants.

	DPPH (IC_50_ µg/mL)
CC	292.02 ± 24.28
AA	9.12 ± 1.91
Trolox	14.22 ± 3.92

CC—*Cotinus coggygria* bark methanol extract; AA—ascorbic acid.

**Table 4 pharmaceutics-17-01293-t004:** Organoleptic characteristics and physical appearance of gel base and CC gel after 1, 60, and 180 days of preparation and storage at 22 ± 2 °C. GB—gel base (placebo gel); CC—gel loaded with *C. coggygria* extract.

Parameter	1 Day	60 Days	180 Days	1 Day	60 Days	180 Days
Color	White	White	White	Brownish	Brownish	Brownish
Odor	/	/	/	Characteristic odor of the extract	Characteristic odor of the extract	Characteristic odor of the extract
Consistency	Semi-solid	Semi-solid	Semi-solid	Semi-solid	Semi-solid	Semi-solid
Homogeneity	No phase separation	No phase separation	No phase separation	No phase separation	No phase separation	No phase separation

**Table 5 pharmaceutics-17-01293-t005:** Histological score analysis of buccal ulcer tissue. GB—gel base; CC—gel loaded with *C. coggygria* extract.

	0 Day	3 Days	6 Days	9 Days
Control	2.00 ± 0.00	2.90 ± 0.15	3.90 ± 0.15	4.90 ± 0.15
Gel base	2.32 ± 0.04	3.3 ± 0.04	4.32 ± 0.04	5.0 ± 0.00
CC gel	2.68 ± 0.04	4.0 ± 0.00	5.0 ± 0.00	NA

NA—not applicable, all animals reached maximum score by day 6.

## Data Availability

The data presented in this study are available in this article.

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
