# Peer review of "Oral Gel Formulation of Cotinus coggygria Scop. Stem Bark Extract: Development, Characterization, and Therapeutic Efficacy in a Rat Model of Aphthous Stomatitis"

_pharmaceutics, 2025, doi:10.3390/pharmaceutics17101293_

Round 1
Reviewer 1 Report
Comments and Suggestions for Authors
This manuscript presents a study on the development, characterization, and pre-clinical evaluation of a novel mucoadhesive oral gel containing methanolic extract of Cotinus coggygria stem bark for the treatment of recurrent aphthous stomatitis. The study reveals comprehensive approach, combining phytochemical analysis of the extract with appropriate physicochemical characterization of the gel, followed by in vivo evaluation in a rat model. The authors demonstrate that the obtained gel possesses excellent stability, significantly accelerates ulcer healing, reduces local oxidative stress, and promotes tissue regeneration, overperforming both an untreated control and a placebo gel base. The work is relevant to the field of pharmaceutical formulation and natural product research for oral mucosal disorders.
After examining the manuscript I have following questions, comments and suggestions:
- The introduction and discussion should incorporate more recent references (2020-2025) to better situate the study within the current state of the art, particularly regarding RAS pathogenesis and mucoadhesive technologies. The reference list is extensive but could be more current. An effort to cite more recent reviews or studies (within the last 3-5 years) on RAS pathogenesis, oxidative stress in oral diseases, and advanced oral mucoadhesive systems would enhance the context of the introduction and discussion.
- Statistics. The description of statistical analysis in the methods is adequate, but its application in the results figures is not transparent. All figure legends (Figure 2, 5, 6) must explicitly state the statistical test used for multiple comparisons and clearly define all the symbols. Legends are incomplete. They must specify the statistical test and the meaning of the symbols used to denote significance.
- Table 6. The histological scoring table is unclear. The group labeled "CC base" is presumably "CC gel". The value for the CC gel group on day 9 is missing ("/"). If the animals were fully healed and sacrificed earlier, this should be stated explicitly ("Not applicable, all animals reached max score by day 6").
- The type of polymer used. The manuscript must be checked for consistency regarding the type of Carbopol used. Methods state 974P, while the abstract and discussion mentions 934. This is a critical detail regarding the formulation composition.
- The absence of a positive control group (for example, triamcinolone acetonide paste) limits the clinical relevance of the findings. Its inclusion would greatly strengthen the impact of the conclusions. This should be acknowledged as a limitation of the study and added or discussed in the work.
Thus, this research is of interest to researchers in pharmaceutics and oral medicine. RAS is a common and debilitating condition with limited treatment options. Research into effective, natural, and safe alternatives to synthetic drugs is a high-priority area. The study presents a stable, effective, and safe prototype formulation that has good translational potential.
My overall recommendation: is that the Major Revisions are required for this work according to reasons above. The manuscript requires revisions to address the critical issues of statistical reporting clarity, data presentation, and terminological consistency before it can be accepted for publication.
Author Response
Reviewer 1
This manuscript presents a study on the development, characterization, and pre-clinical evaluation of a novel mucoadhesive oral gel containing methanolic extract of Cotinus coggygria stem bark for the treatment of recurrent aphthous stomatitis. The study reveals comprehensive approach, combining phytochemical analysis of the extract with appropriate physicochemical characterization of the gel, followed by in vivo evaluation in a rat model. The authors demonstrate that the obtained gel possesses excellent stability, significantly accelerates ulcer healing, reduces local oxidative stress, and promotes tissue regeneration, overperforming both an untreated control and a placebo gel base. The work is relevant to the field of pharmaceutical formulation and natural product research for oral mucosal disorders.
After examining the manuscript I have following questions, comments and suggestions:
Comment 1: The introduction and discussion should incorporate more recent references (2020-2025) to better situate the study within the current state of the art, particularly regarding RAS pathogenesis and mucoadhesive technologies. The reference list is extensive but could be more current. An effort to cite more recent reviews or studies (within the last 3-5 years) on RAS pathogenesis, oxidative stress in oral diseases, and advanced oral mucoadhesive systems would enhance the context of the introduction and discussion.
Response 1: Thank you for your comment. We have updated the manuscript to include more recent references on RAS pathogenesis and on mucoadhesive gels (using the two gelling agents which we used in this study), to better situate our study within the current state of the art. Please see the Introduction, Discussion and Reference sections.
Comment 2: Statistics. The description of statistical analysis in the methods is adequate, but its application in the results figures is not transparent. All figure legends (Figure 2, 5, 6) must explicitly state the statistical test used for multiple comparisons and clearly define all the symbols. Legends are incomplete. They must specify the statistical test and the meaning of the symbols used to denote significance.
Response 2: We thank the reviewer for pointing out the need to specify statistical tests and clarify significance symbols in the figure legends. In the revised manuscript, all figure legends now explicitly state the statistical tests used, depending on data distribution. Additionally, all symbols denoting statistical significance are clearly defined (*p < 0.05). The figure legends have been updated to ensure they are complete and understandable without referring to the main text. Please see Figure legends.
Comment 3: Table 6. The histological scoring table is unclear. The group labeled "CC base" is presumably "CC gel". The value for the CC gel group on day 9 is missing ("/"). If the animals were fully healed and sacrificed earlier, this should be stated explicitly ("Not applicable, all animals reached max score by day 6").
Response 3: Thank you for your comments. This was a technical error, the group previously labeled “CC base” has been corrected to “CC gel.” Regarding the missing value for the CC gel group on day 9, all animals in this group had fully healed and were sacrificed earlier. The table has been updated to indicate this explicitly as “Not applicable, all animals reached maximum score by day 6. Please see Table 5 (the previous Table 6 has been renumbered as Table 5 following the removal of a table)
Comment 4: The type of polymer used. The manuscript must be checked for consistency regarding the type of Carbopol used. Methods state 974P, while the abstract and discussion mentions 934. This is a critical detail regarding the formulation composition.
Response 4: Thank you for pointing this out. The mention of Carbopol 934 in the abstract and discussion was a typographical error. The correct polymer used in the study is Carbopol 974P, and the manuscript has been corrected to reflect this consistently. Please see Abstract and Discussion sections.
Comment 5: The absence of a positive control group (for example, triamcinolone acetonide paste) limits the clinical relevance of the findings. Its inclusion would greatly strengthen the impact of the conclusions. This should be acknowledged as a limitation of the study and added or discussed in the work.
Response 5: Thank you for this useful comment. However, the primary focus of this study was to examine the effects of CC gel itself on buccal ulcer healing. While we acknowledge that including a positive control group (e.g., triamcinolone acetonide or other clinically relevant treatments) could strengthen clinical relevance, this was beyond the scope of the current work. This limitation has been explicitly acknowledged in the revised manuscript, and future studies will focus on comparative evaluations with established treatments. Please see Results and Discussion section, limitation paragraph.
Thus, this research is of interest to researchers in pharmaceutics and oral medicine. RAS is a common and debilitating condition with limited treatment options. Research into effective, natural, and safe alternatives to synthetic drugs is a high-priority area. The study presents a stable, effective, and safe prototype formulation that has good translational potential.
My overall recommendation: is that the Major Revisions are required for this work according to reasons above. The manuscript requires revisions to address the critical issues of statistical reporting clarity, data presentation, and terminological consistency before it can be accepted for publication.
Thank you for your thorough review and constructive comments. We have carefully addressed the critical issues you highlighted, including statistical reporting, clarity of data presentation, and consistency in terminology. We believe that the revised manuscript now reflects these improvements and hope it meets the standards for publication.
Reviewer 2 Report
Comments and Suggestions for Authors
- The manuscript should more clearly explain what is novel compared to previous work on coggygria or other herbal oral gels.
- Please provide a more critical comparison with existing formulations and published wound-healing studies.
- The methanolic extraction method raises concerns about clinical applicability. A justification for the solvent choice and discussion of possible toxic residues are required.
- The antioxidant activity (DPPH ICâ‚…â‚€ ≈ 292 µg/mL) is weak compared to standard antioxidants. This discrepancy should be acknowledged and carefully discussed rather than overstated.
- Include chromatograms and quantitative phytochemical data in supplementary files for transparency.
- Current physicochemical characterization (pH, organoleptic, conductivity, centrifugation) is too basic. Please provide: Rheological analysis (viscosity, gelation temperature); Mucoadhesive strength testing; In vitro release profile of key compounds; Microbiological stability.
- The six-month stability study should be presented with numerical values, statistical analyses, and figures.
- A positive control group (e.g., corticosteroid gel or a marketed preparation) must be included to benchmark efficacy.
- Only one concentration (5% extract) was tested. Dose–response studies are necessary.
- Sample size justification and statistical power analysis should be provided.
- The ulcer area measurement protocol (ImageJ) requires more detail—blinding, calibration, and inter-rater consistency should be specified.
- Safety evaluation is insufficient. Serum chemistry, liver/kidney function markers, or body weight monitoring should be reported, given potential concerns with methanolic extracts.
- Histological scoring should be performed in a blinded manner, with more representative images and ideally quantitative morphometric data.
- The discussion is too strong in claiming “remarkable efficacy” and “safe therapeutic alternative.” Such conclusions are premature without clinical validation.
- Please include a Limitations section acknowledging the single dose, lack of release/safety data, and the preclinical scope of the study.
- Reduce repetition of background information (antioxidant/anti-inflammatory properties of coggygria appear multiple times).
- Ensure all figures/tables have proper error bars, statistical markers, and complete captions.
- English language editing is needed to improve clarity and conciseness.
- In Abstract, avoid overstatement; results should be described more cautiously.
- In Methods, provide details of animal randomization, blinding, and ethical approval reference number.
- For References, some citations are outdated; consider adding recent work on mucoadhesive oral formulations.
English language editing is needed to improve clarity and conciseness.
Author Response
Reviewer 2
Comment 1: The manuscript should more clearly explain what is novel compared to previous work on coggygria or other herbal oral gels.
Response 1: Thank you for this comment. The novelty of our study lies in the unique combination of C. coggygria extract with a synergistic polymer matrix of carbopol® 974P NF and poloxamer 407. To our knowledge, this is the first study to use this specific plant extract with this polymer combination, supported by in vivo validation in aphthous stomatitis rat model. This highlights its distinctive contribution compared to previous studies on C. coggygria extract which primarily focused on extract characterization, in vitro activity, in vivo extract activity in diabetic wounds or other studies on herbal oral gels. Please see the Results and Discussion section.
Comment 2: Please provide a more critical comparison with existing formulations and published wound-healing studies.
Response 2: Thank you for this comment. It was added and discussed in Results and Discussion section. Please Results and Discussion part.
Comment 3: The methanolic extraction method raises concerns about clinical applicability. A justification for the solvent choice and discussion of possible toxic residues are required.
Response 3: We thank the reviewer for their comment and the opportunity to clarify our use of methanolic extraction. We chose methanol as the extraction solvent because it is known to be highly effective at extracting a wide range of polar and semi-polar phytochemicals—especially phenolics and flavonoids, which were the main focus of our phytochemical and bioactivity studies. We understand the concerns about the clinical relevance and possible toxicity of methanol. However, we would like to emphasize that the methanolic extract in our study was thoroughly dried under reduced pressure using a rotary evaporator. After that, it was further dried to a constant weight to ensure that all traces of methanol were completely removed before the extract was used in the in vivo experiments.
Comment 4: The antioxidant activity (DPPH ICâ‚…â‚€ ≈ 292 µg/mL) is weak compared to standard antioxidants. This discrepancy should be acknowledged and carefully discussed rather than overstated.
Response 4: We thank the reviewer for this observation. We agree that the DPPH radical scavenging activity of the methanol bark extract (ICâ‚…â‚€ ≈ 292 µg/mL) is weaker compared with pure standards such as ascorbic acid and Trolox. This difference is expected, as crude extracts represent complex mixtures in which bioactive compounds are diluted by inert phytoconstituents, unlike pure compounds that exhibit higher activity on a molar basis. We have now revised the manuscript to clearly acknowledge this limitation and to avoid overstating the antioxidant potency. Importantly, we emphasize that although the direct radical scavenging effect was modest, the extract demonstrated substantial biological relevance in vivo, where treatment with the CC gel significantly reduced TBARS levels, increased SOD, CAT, and GSH activities, and accelerated ulcer healing. These findings suggest that the therapeutic effects are not solely dependent on direct radical scavenging, but rather on a combination of antioxidant, anti-inflammatory, and regenerative activities of the phytochemicals, particularly sulfuretin. The Discussion section is revised in accordance with the comment above.
Comment 5: Include chromatograms and quantitative phytochemical data in supplementary files for transparency.
Response 5: We thank the reviewer for the suggestion. The chromatograms and quantitative phytochemical data have now been included in the supplementary files to ensure full transparency and reproducibility of the study.
Comment 6: Current physicochemical characterization (pH, organoleptic, conductivity, centrifugation) is too basic. Please provide: Rheological analysis (viscosity, gelation temperature); Mucoadhesive strength testing; In vitro release profile of key compounds; Microbiological stability.
Response 6: We thank the reviewer for the valuable suggestions regarding advanced physicochemical characterization. We have now included rheological analysis to provide additional insight into the formulation properties (please see Results and Discussion). In the present study, our primary focus was on a detailed assessment of the in vivo efficacy of the CC gel in promoting oral ulcer healing. To ensure formulation integrity, we evaluated pH, organoleptic properties, electrical conductivity, centrifugation and rheological analysis. While mucoadhesive strength, in vitro release profiles of bioactive compounds, and microbiological stability are important for comprehensive technological characterization, these analyses were beyond the scope of this efficacy-focused study. We plan to perform these advanced assessments in future studies to complement the current findings and provide a more complete evaluation of the formulation for potential translational and clinical applications.
Comment 7: The six-month stability study should be presented with numerical values, statistical analyses, and figures.
Response 7: We thank the reviewer for the suggestion. The six-month stability study has been revised to include numerical values and statistical analyses. The results are now presented in the manuscript (numerical values within the text), statistical analysis has been added and can be seen in the corresponding figures (please see Figure 2).
Comment 8: A positive control group (e.g., corticosteroid gel or a marketed preparation) must be included to benchmark efficacy.
Response 8: Thank you for this useful comment. However, the primary focus of this study was to examine the effects of CC gel itself on buccal ulcer healing. While we acknowledge that including a positive control group (e.g., triamcinolone acetonide or other clinically relevant treatments) could strengthen clinical relevance, this was beyond the scope of the current work. This limitation has been explicitly acknowledged in the revised manuscript, and future studies will focus on comparative evaluations with established treatments. Please see Results and Discussion section, limitation paragraph.
Comment 9: Only one concentration (5% extract) was tested. Dose–response studies are necessary.
Response 9: We thank the reviewer for this comment. The 5% concentration of C. coggygria extract was selected based on the available literature, where this dose demonstrated significant wound-healing effects in animal model (1). We acknowledge that dose–response studies would provide further insight; however, in accordance with the 3Rs principle (Replacement, Reduction, Refinement) for animal research, we limited our experiments to a single effective dose to avoid unnecessary use of additional animals. Future studies will be designed to systematically evaluate different concentrations while maintaining ethical standards for animal experimentation.
- Aksoy H, Sen A, Sancar M, Sekerler T, Akakin D, Bitis L, Uras F, Kultur S, Izzettin FV. Ethanol extract of Cotinus coggygria leaves accelerates wound healing process in diabetic rats. Pharm Biol. 2016 Nov;54(11):2732-2736.
Comment 10: Sample size justification and statistical power analysis should be provided.
Response 10: Sample size estimation was conducted using G*Power 3 software, guided by previously published studies employing similar methodology (doi: 10.1080/13880209.2016.1181660). The calculation was based on a significance level (α = 0.05) and a statistical power of 0.80 for a two-tailed independent samples t-test comparing groups. This analysis indicated a required total of N = 45 animals, corresponding to 15 rats per group. Details have been provided in the Materials and Methods section, paragraph 2.9.1. Animals.
Comment 11: The ulcer area measurement protocol (ImageJ) requires more detail—blinding, calibration, and inter-rater consistency should be specified.
Response 11: We thank the reviewer for this valuable comment. In the revised manuscript, we have provided a more detailed description of the ulcer area measurement protocol. Specifically, we added that all images were calibrated using a millimeter scale included in each photograph, images were coded and analyzed in a blinded manner, and two experienced investigators independently performed the analysis to ensure consistency. This information has now been integrated into the Methods section (Assessment of the Oral Buccal Ulcer healing potential of CC gel)
Comment 12: Safety evaluation is insufficient. Serum chemistry, liver/kidney function markers, or body weight monitoring should be reported, given potential concerns with methanolic extracts.
Response 12: Animals were monitored daily for signs of pain, distress, or behavioral changes, including locomotor activity, posture, grooming, and food and water intake. Body weight was recorded on days 3, 6, and 10 as an indicator of overall health and systemic toxicity. Any adverse effects at the oral ulcer site or evidence of systemic distress would have prompted immediate intervention. During sacrifice and tissue collection for histological analyses, major organs, including the liver and kidneys, were inspected macroscopically. No visible abnormalities or suspicious changes in size or appearance were observed.
The gel was applied locally to small oral lesions for a short period, which inherently limits systemic absorption. Oral mucoadhesive gels are designed to retain active compounds at the site of application, and at the concentrations used, the main constituents—sulphuretin, fisetin, and hyperoside—are unlikely to cause systemic toxicity or disturbances. Nevertheless, comprehensive safety evaluations, including serum chemistry and organ function markers, are important for assessing long-term safety in humans and will be conducted in future studies. Please see the Results and Discussion section, which has been expanded to address these safety considerations and now includes the body weight results.
Comment 13: Histological scoring should be performed in a blinded manner, with more representative images and ideally quantitative morphometric data.
Response 13. We appreciate the reviewer’s valuable comment and we fully agree that histological scoring should ideally be performed in a blinded manner and supported by more representative images and quantitative morphometric data. However, such an extended histological and morphometric analysis would require additional time and further staining protocols, which unfortunately could not be conducted within the short time available for the review process. The main focus of our work was not an in-depth histological assessment, but rather the development of a novel herbal formulation aimed at accelerating the healing of aphthous stomatitis. Nevertheless, we fully acknowledge the importance of this suggestion and plan to expand our future research with special emphasis on additional histological markers and a more detailed investigation of the mechanisms of action of this plant extract in the aphthous stomatitis model.
Comment 14: The discussion is too strong in claiming “remarkable efficacy” and “safe therapeutic alternative.” Such conclusions are premature without clinical validation.
Response 14: We thank the reviewer for the comment. The discussion and conclusion have been revised to avoid overstatements regarding efficacy and safety. Statements such as “remarkable efficacy” and “safe therapeutic alternative” have been replaced with more cautious phrasing that reflects the preclinical nature of our study. The revised discussion and conclusion now emphasize that the CC gel shows potential for RAS management, while clearly stating that further clinical studies are required to confirm its efficacy and safety in humans. Please see the updated Discussion and Conclusion sections for these changes.
Comment 15: Please include a Limitations section acknowledging the single dose, lack of release/safety data, and the preclinical scope of the study.
Response 15: We have added a Limitations section addressing the points raised; please see the last paragraph of the Results and Discussion section.
Comment 16: Reduce repetition of background information (antioxidant/anti-inflammatory properties of coggygria appear multiple times).
Response 16: Thank you for the comment. We have reduced repetition, please see the revised Abstract, Introduction, Results and Discussion sections.
Comment 17: Ensure all figures/tables have proper error bars, statistical markers, and complete captions.
Response 17: We appreciate the reviewer’s observation. All figures and tables have been revised to include proper error bars, statistical significance markers, and complete, detailed captions specifying the statistical tests used and the meaning of symbols.
Comment 18: English language editing is needed to improve clarity and conciseness.
Response 18: English has now been improved. Please see the Manuscript sections.
Comment 19: In Abstract, avoid overstatement; results should be described more cautiously.
Response 19: Thank you for this comment. The Abstract has been revised to present the results more cautiously, avoiding overstatements and ensuring the conclusions reflect the preclinical scope of the study.
Comment 20: In Methods, provide details of animal randomization, blinding, and ethical approval reference number.
Response 20: We thank the reviewer for the comment. Details regarding animal randomization and blinding have been added to the Methods section (Section 2.9.3), and the ethical approval information has been included in the Institutional Review Board Statement. Please see the revised manuscript.
Comment 21: For References, some citations are outdated; consider adding recent work on mucoadhesive oral formulations.
Response 21: We appreciate the reviewer’s suggestion to include more recent references on mucoadhesive oral formulations. In response, we have updated the manuscript to incorporate recent studies.
Thank you for your thorough review and constructive comments. We have carefully addressed all your comments.
Round 2
Reviewer 1 Report
Comments and Suggestions for Authors
The authors have done a good job in addressing my comments. The revisions are substantial and improved the manuscript's quality, clarity, and scientific rigor. The manuscript is now much stronger and is suitable for publication. The study presents a novel, well-characterized, and effectively evaluated natural formulation with good potential for the treatment of aphthous stomatitis.
So my final decision is that the manuscript can be accepted in present form.
Reviewer 2 Report
Comments and Suggestions for Authors
Authors already revised well.